# Extracellular Vesicles as Mediators of Nickel-Induced Cancer Progression

**DOI:** 10.3390/ijms232416111

**Published:** 2022-12-17

**Authors:** Shan Liu, Angelica Ortiz, Aikaterini Stavrou, Angela R. Talusan, Max Costa

**Affiliations:** Division of Environmental Medicine, Department of Medicine, New York University School of Medicine, New York, NY 10010, USA

**Keywords:** nickel, extracellular vesicles, cancer

## Abstract

Emerging evidence suggests that extracellular vesicles (EVs), which represent a crucial mode of intercellular communication, play important roles in cancer progression by transferring oncogenic materials. Nickel (Ni) has been identified as a human group I carcinogen; however, the underlying mechanisms governing Ni-induced carcinogenesis are still being elucidated. Here, we present data demonstrating that Ni exposure generates EVs that contribute to Ni-mediated carcinogenesis and cancer progression. Human bronchial epithelial (BEAS-2B) cells and human embryonic kidney-293 (HEK293) cells were chronically exposed to Ni to generate Ni-treated cells (Ni-6W), Ni-transformed BEAS-2B cells (Ni-3) and Ni-transformed HEK293 cells (HNi-4). The signatures of EVs isolated from Ni-6W, Ni-3, HNi-4, BEAS-2B, and HEK293 were analyzed. Compared to their respective untreated cells, Ni-6W, Ni-3, and HNi-4 released more EVs. This change in EV release coincided with increased transcription of the EV biogenesis markers CD82, CD63, and flotillin-1 (FLOT). Additionally, EVs from Ni-transformed cells had enriched protein and RNA, a phenotype also observed in other studies characterizing EVs from cancer cells. Interestingly, both epithelial cells and human umbilical vein endothelial (HUVEC) cells showed a preference for taking up Ni-altered EVs compared to EVs released from the untreated cells. Moreover, these Ni-altered EVs induced inflammatory responses in both epithelial and endothelial cells and increased the expression of coagulation markers in endothelial cells. Prolonged treatment of Ni-alerted EVs for two weeks induced the epithelial-to-mesenchymal transition (EMT) in BEAS-2B cells. This study is the first to characterize the effect of Ni on EVs and suggests the potential role of EVs in Ni-induced cancer progression.

## 1. Introduction

Nickel (Ni), a naturally abundant element in the earth’s crust, is widely applied in industrial processes, including the production of batteries, stainless steel, coins, and medical devices, as well as Ni plating, welding, and refining [1,2]. This widespread utilization results in a high degree of occupational and environmental exposure. The toxicity of Ni is affected by the route of exposure, solubility of Ni, dose, and duration of exposure. The inhalation of Ni via the lung is the major route of exposure for Ni-induced toxicity, especially carcinogenicity [3]. Due to the ubiquitous use of Ni, Ni compounds are also found in particulate matter in the ambient environment [4]. In 1990, Ni compounds were categorized as a group I carcinogen by the International Agency for Research (IARC) based on epidemiologic reports and experimental studies on Ni carcinogenesis [5]. However, the mechanisms underlying the carcinogenicity of Ni exposure are still being intensely investigated.

In the past decade, mounting experimental evidence has indicated that extracellular vesicles (EVs) play important roles in cancer development [6,7,8]. EVs are phospholipid bilayer-bound vesicles ranging in size from 30 to 1000 nm that are secreted by cells into the extracellular space and work as intercellular communicators by transferring a variety of biomolecules such as proteins, lipids, and nucleic acids between cells [9]. There are three major types of EVs, microvesicles (MVs), exosomes, and apoptotic bodies. These types of EVs are unique in their biogenesis, release pathway, size, content, and function [10,11,12]. Many cell types such as epithelial cells, T cells, macrophages, reticulocytes, and active neurons can inductively or constitutively secrete EVs to maintain homeostasis or elicit tissue responses [13].

Increasing data indicate that EV signatures, including concentration, size, and contents of EVs, are dynamic, heterogenous, and highly dependent on their cells of origin, environmental conditions, and physiological status [9,14]. Cancer cell-derived EVs contribute to cancer progression by transferring oncogenic materials to cancer cells and non-cancer cells and mediating crosstalk with the tumor microenvironment and pre-metastatic niche [6,15]. Thus, EVs are considered to be one of the most important endogenous carriers of molecular information and are important for cell–cell communication within the tumor microenvironment. However, no study has investigated the effect of Ni exposure on EVs. This study investigates the effects of Ni exposure on EV release and the function of Ni-altered EVs. Following Ni exposure, BEAS-2B and HEK293 released more EVs that are protein- and RNA-enriched compared to untreated cells. These Ni-altered EVs are more easily incorporated by epithelial and endothelial cells compared to EVs from untreated cells. Moreover, EVs from Ni-exposed cells can induce inflammatory responses and the epithelial-to-mesenchymal transition (EMT) in epithelial cells and coagulation and inflammation in endothelial cells. In summary, this is the first study to show that Ni exposure significantly changes EVs’ signatures and functions in epithelial and endothelial cells.

## 2. Results

### 2.1. Ni Exposure Alters EV Signature

To investigate the effect of Ni on EVs, we analyzed the transcriptional changes of EV-related genes identified in the Affymetrix gene expression arrays of seven Ni-transformed clones [16]. The EV-related genes were selected according to the minimal information for studies of extracellular vesicles 2018 (MISEV2018) [17]. In total, 20 EV-related genes were significantly upregulated (*p* < 0.05, fold change (FC) > 1.3) and 10 EV-related genes were significantly downregulated (*p* < 0.05, FC > Ȓ1.3, Figure 1a). Flotillin (FLOT), CD82, and CD63 were selected to examine the mRNA expression levels via qPCR in clone #3 of Ni-transformed BEAS-2B cells (Ni-3) and 6-week Ni-treated BEAS-2B cells (Ni-6W). The mRNA expression levels of FLOT were significantly increased in Ni-6W and Ni-3 cells compared to the controls, which were BEAS-2B cells (Figure 1b). However, the mRNA expression levels of CD82 and CD63 were only elevated in Ni-3 cells, not Ni-6W cells, compared to BEAS-2B cells (Figure 1c,d).

To examine the effects of Ni on EV release, we isolated EVs from the conditioned media of Ni-6W, Ni-3, clone #4 Ni-transformed HEK293 cells (HNi-4), and control BEAS-2B and HEK293 cells. Ni-transformed HEK293 cells were generated by treating HEK293 cells with 100 µM NiCl_2_ for 30 days and selecting single colonies with anchorage-independent growth ability in soft agar (Appendix A). After isolation by serial ultracentrifugation (UC), EVs’ signatures were analyzed using nanoparticle tracking analysis (NTA) (Figure 1e), transmission electron microscopy (TEM) (Figure 1f), and the expression of different EV markers via immunoblotting (Figure 1g). The NTA results show that Ni-treated and Ni-transformed BEAS-2B cells released more EVs compared to the control cells. The concentrations were 5.49 × 10^6^, 2.42 × 10^7^, and 5.16 × 10^7^ particles/mL for control BEAS-2B EVs, Ni-6W EVs, and Ni-3 EVs, respectively (Figure 1e). Compared to HEK293, Ni-transformed HEK293 also released more EVs, with concentrations of 5.65 × 10^11^ and 5.79 × 10^11^ particles/mL, respectively (Appendix A). The average size of EVs from BEAS-2B, Ni-6W, and Ni-3 EVs were 98.5 nm, 145 nm, and 115.6 nm in diameter, respectively (Figure 1e). The average size of EVs from HEK293 and Ni-transformed HEK293 cells were 154.6 nm and 138.7 nm in diameter, respectively (Appendix A). TEM was used to observe the morphology of EVs obtained via ultracentrifugation. As is shown in Figure 1f and Appendix A, the structure of the EVs was intact, and the interior of the EVs was bright white, which indicates that the EVs carried molecules such as proteins and RNAs [10,18]. The expression of the EV markers CD9, TSG101, and flotillin was determined for EVs from all cells (Figure 1g and Appendix A). Total protein and total RNA were extracted from BEAS-2B, Ni-6W, and Ni-3 EVs. Interestingly, Ni-3 EVs had more protein- and RNA-enrichment compared to EVs from BEAS-2B and Ni-6W (Figure 1h). Compared to HEK293 EVs, HNi-4 EVs had significantly higher total protein and RNA concentrations (Appendix A).

### 2.2. EVs from Ni-Treated and Ni-Transformed Cells Increase Uptake in Target Epithelial Cells

EVs can transfer a range of proteins and nucleic acids to recipient cells, and this can affect the phenotype of the recipient cells. We next aimed to examinate if there were any differences in the uptake behavior of EVs from Ni-treated and Ni-transformed epithelial cells compared to EVs from non-exposed cells. EVs from Ni-6W, Ni-3, HNi-4, BEAS-2B, and HEK293 were labeled with lipophilic DiD dye and added to BEAS-2B or HEK293 cells for 1–5 h. Interestingly, the recipient BEAS-2B cells favored the uptake of EVs from Ni-treated and Ni-transformed cells after 3 h (Figure 2a,c) and 5 h (Appendix A) of treatment. An increased number of incorporated EVs was also identified in HNi-4 EVs compared to HEK293 EVs at 1 h (Appendix A). However, no differences were observed between HNi-4 EVs and HEK293 EVs in the 3 h incorporation assay (Appendix A). The uptake of EVs from Ni-6W and Ni-3 was also time-dependent. An increased number of EVs was taken up at 5 h compared to 3 h (Figure 2a,c). To explore whether short-term Ni treatment can affect EV uptake, BEAS-2B cells were pretreated with 100 μM NiCl2 for 24 h before adding EVs to perform the incorporation assays. Interestingly, pretreatment with Ni induced more Ni-6W and Ni-3 EVs to be incorporated at 5 h, but not 3 h (Figure 2e,f). No differences were observed for BEAS-2B EVs in either the 3 h or 5 h incorporation assay (Figure 2d).

### 2.3. EVs from Ni-Transformed Cells Induce Inflammation Response in Target Epithelial Cells

It has been documented that Ni-induced chronic inflammation plays a vital role in its lung carcinogenesis [19,20,21]. We investigated whether EVs from Ni-treated and Ni-transformed epithelial cells could induce inflammatory responses in normal cells. BEAS-2B cells were treated with 10 μg/mL of EVs from Ni-6W, Ni-3, and BEAS-2B cells for 4 h. EVs from BEAS-2B cells had no effect on the mRNA expression levels of inflammatory markers including IL6, MMP1, and CXCL10 (Figure 3a). However, Ni-altered EVs significantly induced the expression of IL6, MMP1, and CXCL10 (Figure 3b,c). To determine the effect of Ni-altered EVs on the EMT process, BEAS-2B cells were treated with Ni-3 EVs for two weeks. The mRNA expression of the epithelial marker E-cadherin (E-CAD) was significantly decreased, while the mRNA expression of the mesenchymal marker N-cadherin (N-CAD) and the master regulator of EMT, ZEB1, were significantly increased (Figure 3d). These results indicate that EVs from Ni-transformed BEAS-2B cells can induce an inflammatory response and EMT in normal BEAS-2B cells. The same results were observed in HEK293 cells treated with EVs from Ni-transformed HEK293 cells (Appendix A).

### 2.4. Ni-Exposure Altered EVs from Epithelial Cells Increase Incorporation in Target Endothelial Cells

Endothelial cells play essential roles in cancer initiation, progression, and metastasis by secreting cytokines, supporting tumor metabolism, and regulating tumor hemostasis and angiogenesis [19]. Previous studies have indicated that cancer cells can affect surrounding endothelial cells by transporting oncogenic factors via EVs [20,21]. In the present study, human umbilical vein endothelial cells (HUVEC) were used to assess the effect of EVs from Ni-exposed epithelial cells on endothelial cells. Under normal conditions, HUVEC cells can incorporate EVs released from normal epithelial cells (Figure 4a). However, as with epithelial cells, endothelial cells showed increasing uptake of EVs from Ni-treated and Ni-transformed BEAS-2B (Figure 4a,b). Interestingly, compared to EVs from un-treated BEAS-2B cells, EVs from HKE293 showed increased incorporation of HUVEC cells (Appendix A). However, the incorporation of HNi-4 EVs was higher than that of HEK293 EVs (Appendix A).

### 2.5. EVs from Ni-Exposed Epithelial Cells Regulate Inflammation and Coagulation in Endothelial Cells

We further anticipated that EVs from Ni-exposed epithelial cells were functional in endothelial cells. HUVEC cells were treated with EVs from Ni-6W, Ni-3, and untreated BEAS-2B for 4 h. The mRNA expression levels of the coagulation marker F5 and inflammation markers, including matrix metalloproteinase-9 (MMP9), tyrosine kinase (TEK/TIE2), and vimentin (VIM), were measured after EV treatment. EVs from Ni-6W and Ni-3 significantly induced the mRNA expression of F5, MMP9, TEK, and VIM (Figure 5a–d). EVs from BEAS-2B cells did not change the mRNA expression levels of coagulation markers in HUVEC cells (Figure 5a–d). EVs from HNi-4 cells significantly increased the mRNA expression levels of MMP9 and TEK (Appendix A). 

## 3. Discussion

In the past decade, great efforts have been made to elucidate EV biology and the function of EVs in cell-to-cell communication and its contribution to human diseases [9]. Strong evidence has indicated that EVs are crucial in cancer, from regulating carcinogenesis to tumorigenesis and metastasis, as they transfer oncogenic materials and modify the tumor microenvironment [6,7,8]. The carcinogenicity of Ni has been a health concern for a long time, but the underlying mechanisms have not been fully described. This is the first study to show the effect of Ni on the biogenesis, incorporation, and function of extracellular vesicles, which paves the way for the study of Ni-mediated cancer progression.

In the present study, 30 EV-related genes were differentially expressed in the Affymetrix gene expression arrays of seven Ni-transformed clones. We further verified that two tetraspanin genes, CD82 and CD63, and flotillin were significantly increased in Ni-transformed BEAS-2B cells compared to Ni-treated BEAS-2B and untreated cells. Although the CD63 level was not increased according to the Affymetrix gene expression arrays of a mixture of 7 Ni-transformed clones, it was significantly increased in the Ni-transformed clone #3 used in this study. Tetraspanins, a family of integral membrane proteins located on plasma membrane and intracellular membrane structures, such as endosomal or lysosomal compartments, play important roles in regulating EV biogenesis, cargo selection, and cell targeting [22]. Flotillin is a component of lipid rafts and a positive regulator of endocytosis [23]. Ni-treated and Ni-transformed cells showed increased EV release compared to normal BEAS-2B cells. The elevated expression of CD82, CD63, and flotillin in Ni-exposed cells partially explains the increased EV release from Ni-treated and Ni-transformed cells. Increased EV release was also observed in Ni-transformed HEK293 cells. The total protein and total RNA concentrations were also increased in EVs from Ni-transformed BEAS-2B cells compared to untreated cells. These results align with previous reports of elevated EV release in cancer cells compared to non-malignant cells [24,25,26]. The potential factors that regulate EV release in cancer cells include hypoxia, endoplasmic reticulum stress, autophagy, oxidative stress, and an increase in intracellular calcium ion levels [27,28,29,30,31]. Prior studies have reported that Ni exposure can activate the hypoxia signaling pathway, induce ER stress and oxidative stress, and elevate Ca^2+^ levels [32,33,34].

EVs can transfer a range of proteins and nucleic acids to recipient cells and can affect the phenotype of recipient cells. The first step of this process is the entry of EVs into the target cells. EVs can be incorporated by recipient cells via multiple pathways, including ligand–receptor binding, endocytosis, phagocytosis, and surface membrane fusion [13,35]. EV uptake has not been fully studied. However, EV uptake generally involves a combination of different pathways, and it seems to depend on the component and origin of the EVs [10,35]. In the present study, BEAS-2B cells showed an increased uptake of EVs released from Ni-treated and Ni-transformed BEAS-2B in a time-dependent manner. Pre-treatment with Ni in BEAS-2B cells resulted in a further elevation in the number of Ni-exposed EVs being incorporated by BEAS-2B cells compared to cells without Ni pretreatment. However, no difference was observed in the BEAS-2B EV treatment groups. In our results, most of the EVs incorporated by the cells were colocalized with the plasma membrane, and some EVs were colocalized with the nucleus. Few EV signals stayed red in the merged channels, which indicates that the EVs were docking or undergoing phagocytosis [36,37]. These results indicate that the increase in EV uptake induced by Ni exposure was dependent on the components of the EVs but not the status of the recipient cells.

Previous studies suggest that chronic lung inflammation is responsible for lung cancer induced by Ni exposure. Ni exposure stimulates the release of inflammatory factors via the activation of mitogen-activated protein kinases (MAPKs), nuclear factor kappa B (NF-κB), interferon regulatory factor 3 (IRF3), and the Nod-like receptor 3 (NLRP3) inflammasome pathway [38,39,40]. In the first and the second parts of this study, we identified that Ni exposure can modify the EV release, content, and uptake. The next important question is whether these Ni-altered EVs are functional in the recipient cells. According to our results, the transcription levels of the inflammatory markers IL6, MMP1, and CXCL10 were significantly increased in epithelial cells following treatment with EVs derived from Ni-transformed cells. The sustained inflammatory microenvironment formed by Ni induces the activation of autophagy and the release of inflammatory markers, eventually contributing to BEAS-2B cells’ transformation [41]. Thus, our results suggest that EVs may play an important role in maintaining the inflammatory microenvironment following Ni exposure.

In addition to being an inflammatory marker, IL6 is also an inducer of the epithelial–mesenchymal transition (EMT) [42]. EMT is a process by which epithelial cells gain the features of mesenchymal cells. In cancer, the EMT process is associated with carcinogenesis, tumorigenesis, and metastasis [43]. It has been demonstrated that Ni exposure stimulates EMT through the epigenetic activation of ZEB1 in BEAS-2B cells. In the present study, 2 weeks of treatment of Ni-3 EVs induced EMT in BEAS-2B cells. This result is consistent with that of previous studies that showed that EVs from cancer cells can induce EMT in recipient cells by delivering EMT regulators [44].

The tumor microenvironment, which is formed by cancer cells, stromal cells, signaling molecules, and the extracellular matrix, is important in cancer cell progression and metastasis [45]. EVs are present in the tumor microenvironment and transfer signals between the cells [46]. Thus, it is also important to know whether EVs from Ni-treated and Ni-transformed epithelial cells can be incorporated by stromal cells, such as endothelial cells, and have the capability to regulate cell signaling in the recipient cells. According to our results, EVs from Ni-altered BEAS-2B cells can be successfully incorporated by human umbilical vein endothelial cells. Similar to the results from the incorporation assay in epithelial cells, endothelial cells also exhibited uptake of EVs from Ni-treated and Ni-transformed cells compared to EVs from normal BEAS-2B cells. The accumulation of coagulation at cancer sites prevents cancer cells from being recognized by the immune system and aids cancer cell migration [6,47]. We further found that EVs from Ni-treated and Ni-transformed BEAS-2B cells can induce the transcriptional expression of the coagulation maker F5 and inflammation markers MMP9, TEK, and VIM in HUVEC cells. F5 is an essential regulator in the blood coagulation cascade, and F5 expression has been linked to increased cancer risk and tumor aggressiveness in breast cancer, colorectal cancer, and gastric cancer [48,49,50]. TEK or TIE2 is an endothelial cell-specific transmembrane receptor kinase and is essential for vascular remodeling [51]. VIM is a type III intermediate filament protein that is expressed in endothelial cells and regulates their migration, invasion, and vascularization [52]. VIM promotes inflammation via NLRP3 inflammasome activation [53]. Matrix metalloproteinases participate in modeling the extracellular matrix, tumor neovascularization, and metastasis [54]. MMP activities involved in inflammatory responses are essential for vascular remodeling. MM9 expression can be activated by cytokines via NF-κB and activator protein 1 (AP-1) signaling activation [55,56]. Thus, our results suggest that EVs from Ni-exposed epithelial cells are also functional in endothelial cells.

## 4. Materials and Methods

### 4.1. Cell Culture and Ni Treatments

Human bronchial epithelial cells BEAS-2B were obtained from ATCC (CRL-9609, ATCC, Manassas, VA, USA). BEAS-2B cells were cultured in Bronchial Epithelial Cell Growth Medium (BEGM, Lonza, Walkersville, MD, USA) supplemented with 1% penicillin–streptomycin (GIBCO, Grand Island, NY, USA). Ni-transformed BEAS-2B cells and Ni-treated BEAS-2B cells were previously generated and characterized [16,57]. In brief, the Ni-transformed BEAS-2B cells (Ni-3) were generated by exposing BEAS-2B cells to 250 µM soluble nickel (NiSO_4_) for 4 weeks, and transformed clones were selected based on their growth in soft agar. Ni-treated BEAS-2B cells (Ni-6W), a generous gift from Dr. Cuddapah (New York University School of Medicine, New York, NY, USA), were generated by treating BEAS-2B cells with 100 µM NiCl_2_ for 6 weeks. Human embryonic kidney-293 cells (HEK293, CRL-1573.3, ATCC) were cultured in Dulbecco’s Modified Eagle’s Medium (DMEM, Invitrogen, Grand Island, NY, USA) with 10% heat-inactivated fetal bovine serum (FBS, GIBCO, Grand Island, NY, USA) and 1% penicillin–streptomycin. HEK293 cells were treated with 100 μM NiCl_2_ (Sigma Aldrich, St. Louis, MO, USA) for 4 weeks, after which transformed clones were selected from soft agar as previously described [16]. Human umbilical vein endothelial cells (HUVEC, CRL-1730, ATCC) were cultured in 0.2% gelatin-coated plates supplemented with 15% FBS, 1% P/S, and 1% endothelial cell growth factor. All cells were cultured in 37 °C incubators with 5% CO_2_. EV-free FBS was used in the cell culture medium for all cells used for isolating EVs, EV incorporation assay, and EV treatment.

### 4.2. Isolation of Extracellular Vesicles

Extracellular vesicles were isolated by a standard serial ultra-centrifugation method as described previously [58]. Briefly, cells were cultured in 150 mm dishes, and cell culture medium was harvested when the cells reached 80% confluence. This medium was centrifuged at 2000× *g* for 15 min followed by 10,000× *g* for 15 min at 4 °C to remove dead cells and cell debris. The supernatant medium was then centrifuged at 25,000× *g* overnight at 4 °C. The EV pellet was resuspended and washed in PBS (Invitrogen, Grand Island, NY, USA) by centrifuging for another round. Then, we discarded the supernatant and resuspended the EV pellet in PBS. EVs were stored at −80 °C for long-term storage and thawed on ice before use.

### 4.3. Characterization of Extracellular Vesicles

The isolated extracellular vesicles were visualized via transmission electron microscopy (TEM) by the Core of Microscopy Laboratory in the New York University School of Medicine. The number, size distribution, and concentration of EVs were analyzed using Zetaview QUATT (Particle Metrix, Inning am Ammersee, Germany). The total protein concentration of EVs was quantified using a Pierce™ BCA protein assay kit (Thermo Fisher Scientific, Waltham, MA, USA) according to the manufacturer’s instructions. Total RNA was isolated using a total exosome RNA and protein isolation kit (Thermo Fisher Scientific, Waltham, MA, USA). The purity and quantity of the total RNA extracted from each EV sample were immediately measured via UV absorbance spectroscopy on a NanoDrop 2000 spectrophotometer system (Thermo Fisher Scientific, Waltham, MA, USA).

### 4.4. Extracellular Vesicles Incorporation Assays

EVs were labeled with 5 µL of the lipophilic dye DiD cell-labeling solution (Invitrogen, Grand Island, NY, USA) dissolved in 100% ethanol, incubated at 37 °C for 20 min, transferred into ultracentrifugation tubes, washed with 12 mL of PBS, and resuspended in PBS. BEAS-2B, HEK293, and HUVEC cells were seeded 1 × 10^5^ on a 0.17 mm, #1.5 thickness cover slip (Thomas Scientific, Swedesboro, NJ, USA) in 24-well plates. The cells were pre-treated with PBS or 100 µM NiCl_2_ for 24 h. The protein concentration of the DiD-labeled EVs were measured using a Pierce™ BCA protein assay kit. Then, 5 µg/mL of DiD-labeled EVs were added to cells and incubated at 37 °C for 1–5 h (HEK293 cells: 1 h, BEAS-2B cells: 3 h and 5 h, HUVEC cells: 5 h). After incubation, the cells were washed three times with PBS, fixed by 100% methanal for 20 min at −20 °C, incubated with wheat germ agglutinin, alexa fluor™ 488 conjugate (WGA, Invitrogen, Grand Island, NY, USA) for 10 min, permeabilized by 0.1% triton-100 for 5 min, and washed three times with PBS. We transferred each cover slip to a slide with 15 µL of Molecular Probes™ SlowFade™ diamond antifade mountant with DAPI and kept it in the dark for drying for 10 min. Further, the slides were analyzed using immunofluorescent microscopy (ZOE Fluorescent Cell Imager, Bio-Rad, Berkeley, CA, USA) and confocal microscopy (LSM880 Zeiss Microscope, Zeiss, Jena, Germany).

### 4.5. Western Blot for EV Markers

The EV pellets obtained from ultracentrifugation and cell pellets were lysed with 30 µL of boiling buffer with protease inhibitor. The protein concentrations were measured using a BCA protein assay kit. A total of 100 µg of protein lysates were mixed with 6× sample buffer and boiled at 95 °C for 10 min and then loaded into a gel (Novex WedgeWell 8 to 16%, Tris-Glycine, 1.0 mm, Mini Protein Gel, 12-well, Thermo Fisher Scientific, Waltham, MA, USA). The protein was transferred into a Pierce PVDF Transfer Membrane, 0.45 µm (Thermo Fisher Scientific, Waltham, MA, USA), blocked with 5% milk (Blotting-Grade Blocker, Bio-Rad, Berkeley, CA, USA) in TBST, and then incubated in the following primary antibodies overnight at 4 °C: anti-flotillin, anti-TSG101, anti-CD9, and anti-GAPDH (Cell Signaling Technology, Danvers, MA, USA). After being washed with TBST for 10 min three times, the membranes were incubated in the AP-conjugated secondary antibodies (Cell Signaling Technology, Danvers, MA, USA) for 3 h at 4 °C, washed three times, and exposed to Cytiva Amersham™ ECF™ Substrate (Thermo Fisher Scientific, Waltham, MA, USA) before visualizing the membranes using a Typhoon FLA 7000 biomolecular imager (GE Healthcare, Chicago, IL, USA).

### 4.6. RNA Isolation and qPCR

BEAS-2B, HEK293, and HUVEC cells were seeded in 6-well plates and treated with PBS or EVs for 4 h. After 4 h, the cells were washed with cold PBS three times, collected in Tri reagent, and either processed immediately for RNA isolation or stored at −80 °C. The purity and concentration of each RNA sample were measured using a NanoDrop 2000 spectrophotometer system. Reverse transcription was performed using a high-capacity RNA-to-cDNA (Applied Biosystems, Waltham, MA, USA) Kit with 1000 ng of RNA in a final volume of 10 µL. The mRNA expression levels were measured by quantitative real-time PCR using SYBR green master mix (Thermo Fisher Scientific, Waltham, MA, USA). The primers used are listed in Table 1.

### 4.7. Graphical Depictions and Statistical Analyses

Statistical analyses were performed using GraphPad Prism 9 (San Diego, CA, USA). Differences were assessed by Student’s t test or analysis of variance (ANOVA), depending on the comparison groups. All values are displayed as mean values ± SEM. Differences were considered statistically significant at * *p* < 0.05, ** *p* < 0.01, *** *p* < 0.001, and **** *p* < 0.0001, as shown in the figures.

## 5. Conclusions

Herein, for the first time, we have identified EVs as a potential mediator of Ni-induced cancer progression. EV signatures, including size, number, contents, and uptake, were studied in depth in this study. Ni exposure was able to increase the release of protein- and RNA-enriched EVs, and these EVs were preferably incorporated by epithelial and endothelial cells. Genes that controlled EV biogenesis and content were highly expressed in the Ni-treated and Ni-transformed cells. Moreover, these Ni-altered EVs are functional in both epithelial and endothelial cells and regulate inflammation, EMT, and coagulation, which indicates the potential carcinogenicity of these Ni-altered EVs.

## Figures and Tables

**Figure 1 ijms-23-16111-f001:**
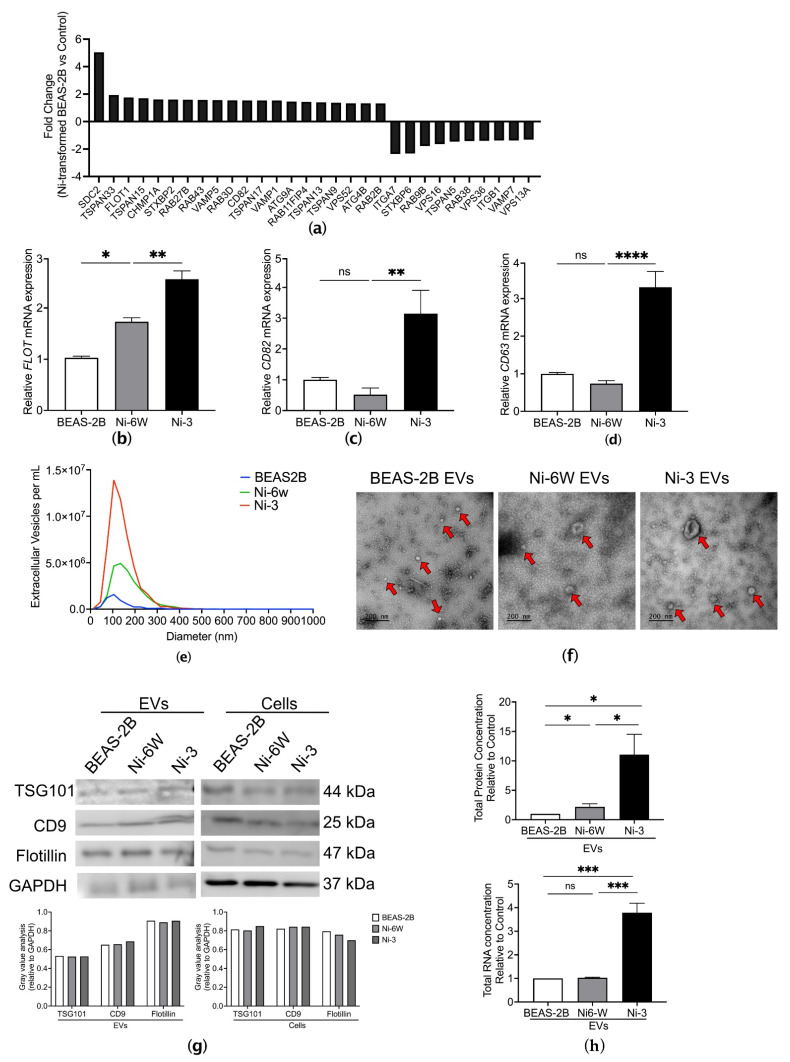
Characterization of EVs from Ni-treated and Ni-transformed BEAS-2B cells. (**a**) A total of 20 upregulated EV-related genes and 10 downregulated EV-related genes were identified in the Affymetrix gene expression arrays of Ni-transformed clones (*p* < 0.05, FC > 1.3). (**b**–**d**) Comparison of mRNA expression of FLOT, CD82, and CD63 in BEAS-2B, Ni-6W, and Ni-3 cells. EVs from BEAS-2B, 6-week-Ni-treated BEAS-2B (Ni-6W), and Ni-transformed BEAS-2B (Ni-3) were subjected to serial ultracentrifugation (UC) and characterized by several methodologies including (**e**) nanoparticle tracking analysis (NTA), (**f**) transmission electronic microscopy (TEM), and (**g**) immunoblot of EV markers CD9, flotillin, and TSG 101. Red arrow: extracellular vesicles. (**h**) Comparison of total protein and total RNA of EVs between Ni-treated, Ni-transformed, and untreated BEAS-2B cells. Values are presented as mean ± SEM. ns: no significance, * *p* < 0.05, ** *p* < 0.01, *** *p* < 0.001, and **** *p* < 0.0001.

**Figure 2 ijms-23-16111-f002:**
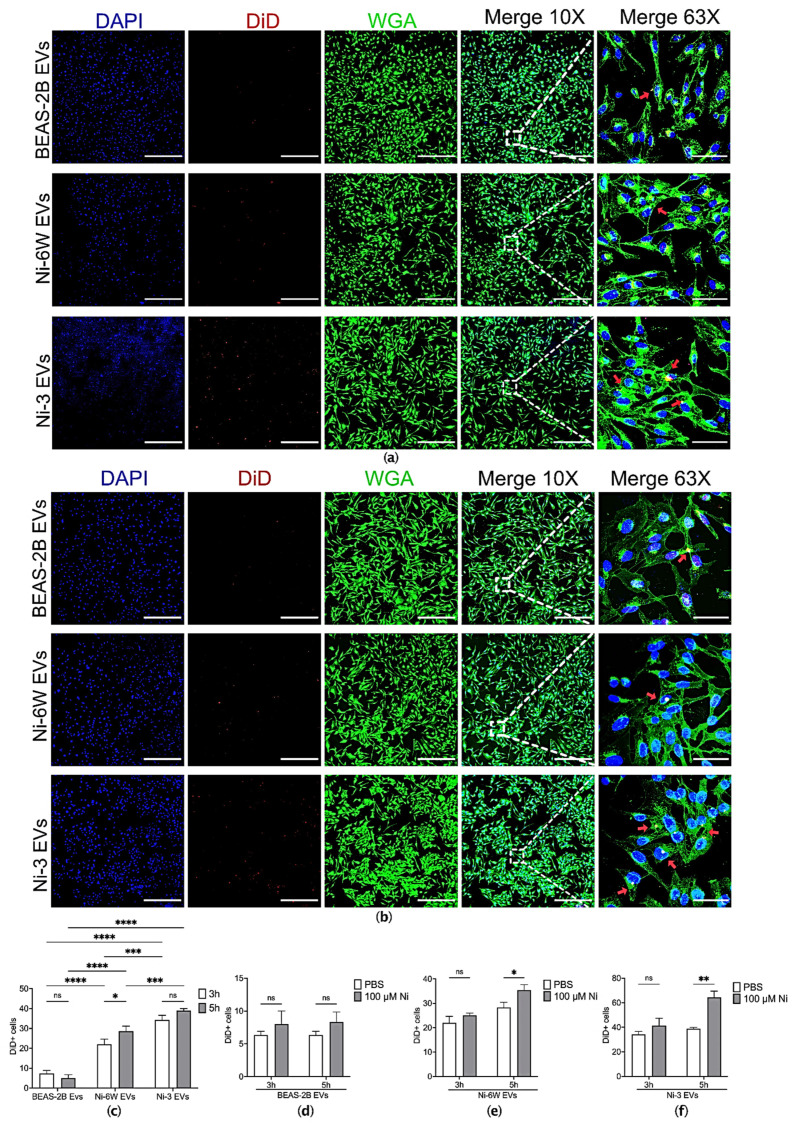
The uptake of EVs was affected by Ni exposure in BEAS-2B cells. (**a**) EVs from Ni-6W, Ni-3, and BEAS-2B were isolated, labelled with DiD dye, and added to BEAS-2B cells for 3 h before being scanned with a confocal microscope (scale bar: 400 μm (10×) and 60 μm (63×), Zeiss 880). (**b**) BEAS-2B cells were pretreated with 100 μM Ni for 24 h before performing a 3 h EV-incorporation assay. Blue—DAPI, red—DiD, green—WGA, red arrow—incorporated EVs. Quantification and comparisons of the number of (**c**) incorporated EVs in the non-Ni-pretreated group, (**d**) BEAS-2B EVs in the Ni-pretreated and untreated groups, (**e**) Ni-6W EVs in the Ni-pretreated and untreated groups, and (**f**) Ni-3 EVs in the Ni-pretreated and untreated groups. Values are presented as mean ± SEM. ns: no significance, * *p* < 0.05, ** *p* < 0.01, *** *p* < 0.001, and **** *p* < 0.0001.

**Figure 3 ijms-23-16111-f003:**
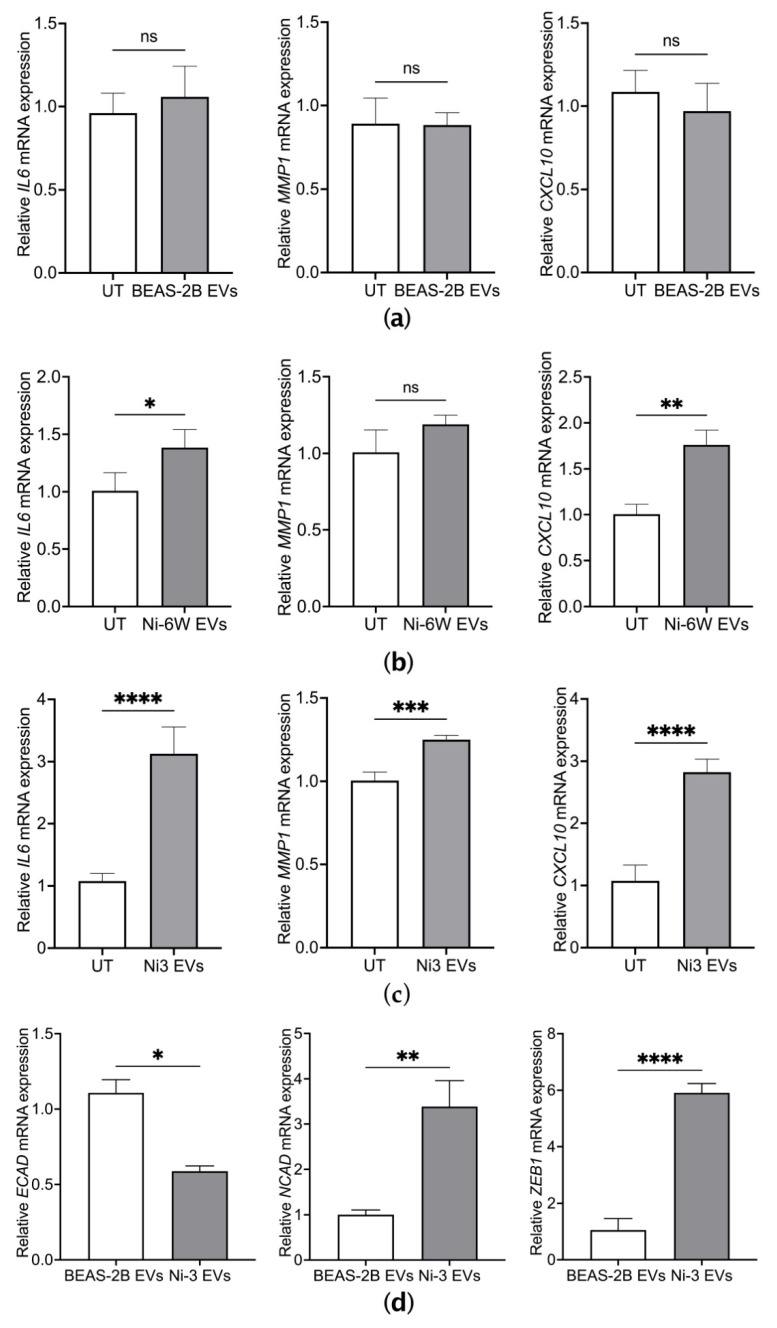
EVs from Ni-transformed cells induced the mRNA expression of inflammatory markers and EMT markers in BEAS-2B cells. The mRNA expression of inflammatory markers including IL6, MMP1, and CXCL10 were measured by quantitative real-time PCR after BEAS-2B cells were treated for 4 h with 10 μg/mL of EVs from BEAS-2B (**a**), Ni-6W (**b**), and Ni-3 cells (**c**). The mRNA expression levels of EMT markers were measured after 2 weeks of treatment with 5 μg/mL of Ni-3 EVs in BEAS-2B cells (**d**). Values are presented as mean ± SEM. ns: no significance, * *p* < 0.05, ** *p* < 0.01, *** *p* < 0.001, and **** *p* < 0.0001.

**Figure 4 ijms-23-16111-f004:**
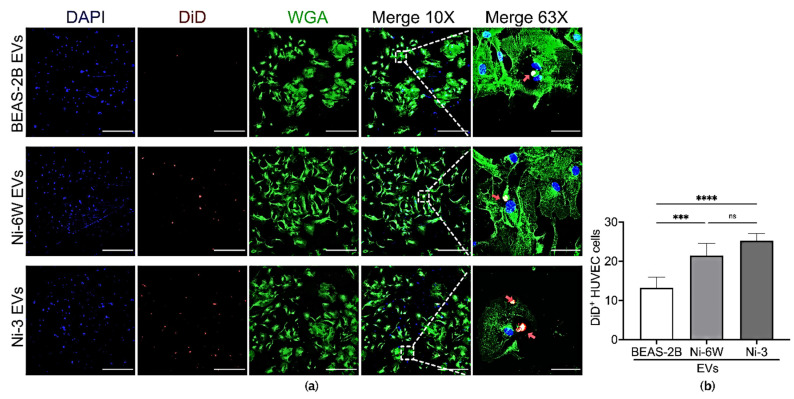
Ni exposure affected the incorporation of BEAS-2B-derived EVs in HUVEC cells. (**a**) EVs from BEAS-2B, Ni-6W, and Ni-3 were isolated, labelled with DiD dye, and added to HUVEC cells for 5 h before being scanned with a confocal microscope (scale bar: 400 μm (10×) and 60 μm (63×), Zeiss 880). Blue—DAPI, red—DiD, green—WGA, red arrow—incorporated EVs. (**b**) Quantification and comparisons of the number of incorporated EVs in HUVEC cells. Values are presented as mean ± SEM. ns: no significance, *** *p* < 0.001 and **** *p* < 0.0001.

**Figure 5 ijms-23-16111-f005:**
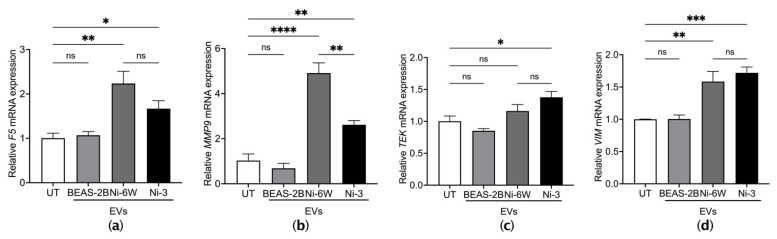
EVs from Ni-exposed BEAS-2B cells altered coagulation and inflammation markers in HUVEC cells. HUVEC cells were treated with 10 μg/mL of EVs from BEAS-2B, Ni-6W, and Ni-3 cells. The mRNA expression of the coagulation marker (**a**) F5 and inflammation markers (**b**) MMP9, (**c**) TEK, and (**d**) VIM were measured by quantitative real-time PCR. Values are presented as mean ± SEM. ns: no significance, * *p* < 0.05, ** *p* < 0.01, *** *p* < 0.001, and **** *p* < 0.0001.

**Table 1 ijms-23-16111-t001:** Primer pairs for qPCR.

Gene	Forward Primer (5′ ->3′)	Reverse Primer (5′ ->3′)
CD63	ATG ATC ACG TTT GCC ATC TT	AGG GAT TTT CTC CCA ATC TG
CD82	AAAGCAGAACCCGCAGAGTC	AAGACATAGGCCCCCATCCT
CXCL10	TTCCTGCAAGCCAATTTTGTC	TCTTCTCACCCTTCTTTTTCATTGT
FLOT	CTCTCATCTCTGCGGTCAGT	TCAACCTCGGCTACTTCTTG
GAPDH	AGGGCTGCTTTTAACTCTGGT	CCCCACTTGATTTTGGAGGGA
IL6	GGTACATCCTCGACGGCATCT	GTGCCTCTTTGCTGCTTTCAC
MMP1	GAGATCATCGGGACAACTCTCCTT	GTTGGTCCACCTTTCATCTTCATCA
MMP9	ATCCAGTTTGGTGTCGCGGAGC	GAAGGGGAAGACGCACAGCT
TEK	CTATCGGACTCCCTCCTCCAA	TCAAATTTAGAGCTGTCTGGCTTTT
F5	CCAGGTAGCTGGCATGCAA	CCATTGGCATCTTACACTCTTTGT
VIM	GACGCCATCAACACCGAGTT	CTTTGTCGTTGGTTAGCTGGT
ZEB1	AGCAGTGAAAGAGAAGGGAATGC	GGTCCTCTTCAGGTGCCTCAG
E-CAD	TGCCCAGAAAATGAAAAAGG	GTGTATGTGGCAATGCGTTC
N-CAD	TGGGAATCCGACGAATGG	TGCAGATCGGACCGGATACT

## Data Availability

Not applicable.

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
