# Peer review of "Extracellular Vesicles as Mediators of Nickel-Induced Cancer Progression"

_ijms, 2022, doi:10.3390/ijms232416111_

Round 1

Reviewer 1 Report

In this work, authors reported nickel exposure can result in increased release of EVs as well as the increased transcription of several EV markers. EVs from Ni-treated cells also was shown more easily incorporated by both epithelial and endothelial cells. This work is potentially interesting as the first time reporting an association between nickel and EVs in cancer progression. There are several issues needed to be addressed.

1.    At line 88, authored reported the bright white indicates EV carrying molecules. Author needs to provide either previously reported reference or direct evidences.

2.    In Figure 1C, molecular weight markers are needed for the WB gel.

3.    Author reported increased proteins and RNA in Ni-transformed cells, however, these are actually not represented in Figure 1C WB results as no significant increase (even decrease in Flotillin and GAPDH) on EVs in Ni-3.  Can authors justify them?

4.    Scale bars for Figure2a/b and Figure 4a are missing.

5.    It is known that the conventional microscopy like confocal has the maximal resolution at ~ 250nm due to the diffraction limit, therefore the direct observation of EVs is very tricky. Meanwhile, author reported the average diameter of EVs are 98.5nm, 145nm and 115.6 nm for three analyzed cells. Additionally, these DiD signal are not in the cells or fused with WGA membrane signal. Therefore, these red particles in Figure 2a/2b or Figure 4 might not be EVs. Author needs to provide more evidences.

Reviewer 2 Report

In this manuscript, authors are the first to report the effect of Ni on EVs and suggest the potential role of EVs in inflammation and cancer progression. This reviewer found that the approaches used in this study is intriguing and will be a great interest to the readers of International Journal of Molecular Sciences if some of the minor corrections and concerns are clarified.

1. It seems better to clearly distinguish the results and discussion part as there are many references included in the results part, which could be removed to discussion or introduction part.

2. Based on the study, even though authors showed that Ni-induced EVs can induce inflammation and thrombotic signals in epithelial and endothelial cells respectively, there seems no direct evidence that these EVs could potentially affect cancer progression. Can author provide proper rationale whether the expression, 'nickel-induced cancer progression' in the title is appropriate to cover the whole study?

3. There are minor spelling and spacing errors must be corrected before publication. For example, in line 130, 'inlammatory responses' -> 'inflammatory responses'. 'whether  EVs from' -> 'whether EVs from' (unnecessary space). 

Round 2

Reviewer 1 Report

Scale bars in Figure 4 are still missing.

Author Response

Dear Reviewer, 

Thank you very much for your comment. We do add the scale bars in Figure 4 in the Word version, which shows the tracked changes. However, the PDF version, which was automatically generated by the system, has a different format. Thus, the missing scale bar is a format issue.

We uploaded a clean version here, please look at this version. Thank you!
